# Ultrastructural and Functional Characterization of Mitochondrial Dynamics Induced by Human Respiratory Syncytial Virus Infection in HEp-2 Cells

**DOI:** 10.3390/v15071518

**Published:** 2023-07-07

**Authors:** Ignacio Lara-Hernandez, Juan Carlos Muñoz-Escalante, Sofía Bernal-Silva, Daniel E. Noyola, Rosa María Wong-Chew, Andreu Comas-García, Mauricio Comas-Garcia

**Affiliations:** 1High-Resolution Microscopy Section, Center for Research in Health Sciences and Biomedicine, Autonomous University of San Luis Potosí, San Luis Potosí 78210, Mexico; lara.ignacioh@gmail.com; 2Center for Research in Health Sciences and Biomedicine, Autonomous University of San Luis Potosí, San Luis Potosí 78210, Mexico; carlos.escalante@uaslp.mx (J.C.M.-E.); dnoyola@uaslp.mx (D.E.N.); 3Department of Microbiology, School of Medicine, Autonomous University of San Luis Potosí, San Luis Potosí 78210, Mexico; sofia.bernal@uaslp.mx; 4Genomic Medicine Section, Center for Research in Health Sciences and Biomedicine, Autonomous University of San Luis Potosí, San Luis Potosí 78210, Mexico; 5Research Division, School of Medicine, National Autonomous University of Mexico, Mexico City 04360, Mexico; rmwongch@yahoo.com.mx; 6Science Department, Autonomous University of San Luis Potosí, San Luis Potosí 78210, Mexico; 7Molecular and Translation Medicine Section, Center for Research in Health Sciences and Biomedicine, Autonomous University of San Luis Potosí, San Luis Potosí 78210, Mexico

**Keywords:** human respiratory syncytial virus, mitochondrial alterations, ultrastructural characterization

## Abstract

Human respiratory syncytial virus (hRSV) is the leading cause of acute lower respiratory tract infections in children under five years of age and older adults worldwide. During hRSV infection, host cells undergo changes in endomembrane organelles, including mitochondria. This organelle is responsible for energy production in the cell and plays an important role in the antiviral response. The present study focuses on characterizing the ultrastructural and functional changes during hRSV infection using thin-section transmission electron microscopy and RT-qPCR. Here we report that hRSV infection alters mitochondrial morphodynamics by regulating the expression of key genes in the antiviral response process, such as Mfn1, VDAC2, and PINK1. Our results suggest that hRSV alters mitochondrial morphology during infection, producing a mitochondrial phenotype with shortened cristae, swollen matrix, and damaged membrane. We also observed that hRSV infection modulates the expression of the aforementioned genes, possibly as an evasion mechanism in the face of cellular antiviral response. Taken together, these results advance our knowledge of the ultrastructural alterations associated with hRSV infection and might guide future therapeutic efforts to develop effective antiviral drugs for hRSV treatment.

## 1. Introduction

The human respiratory syncytial virus (hRSV), also known as human Orthopneumovirus, is an enveloped, single-stranded negative-sense RNA virus that encodes for 11 proteins. This virus belongs to the genus *Orthopneumovirus* and *Pneumoviridae* family [1]. Since it was discovered hRSV has been considered the main etiological agent of acute lower respiratory tract infections in children under the age of five and an important cause of pneumonia in elderly adults worldwide. In 2019, it was estimated that there were 33.0 million acute lower respiratory tract infections related to hRSV infections globally in children 0 to 60 months of age, of which 6.6 million were reported in infants < 6 months [2]. This virus is classified based on its antigenic reactivity to monoclonal antibodies directed against the viral attachment protein (G), and it is further divided into two subgroups: A and B [3]. The hRSV virion consists of viral envelope which has three viral transmembrane proteins: attachment (G), fusion (F), and the small hydrophobic (SH). The inner part of the virion contains the matrix (M) protein which surrounds the nucleoprotein complex. This complex consists of a 15 kb-long genomic RNA, the nucleoprotein (N), the phosphoprotein (P), the large polymerase subunit (L), and the M2-1/2 proteins [4].

Glycoprotein G is of particular interest because it has a larger variability with respect to the other hRSV proteins [5]. Based on this variability, new genotypes have been identified; in particular, duplications of the G protein’s C-terminal domain have resulted in highly dominating genotypes. The first duplication found in hRSV-B was reported in 2003 and consisted of 60 nucleotides (nts) of the C-terminal (genotype BA), but it was determined that the duplication occurred near 1996 [6,7]. In 2012 a second duplication (72 nts) in this gene was reported on an hRSV subgroup A (genotype ON1) [8]. Later it was published that this duplication emerged in multiple locations around the world since 2009 [9] Since their identification, both the BA and ON1 genotypes have become the dominant genotypes worldwide [10]. However, it is not clear why genotypes with these duplications have resulted in replacement of all other variants.

The infectious viral cycle results in remarkable alterations of the cellular architecture. For example, certain viruses have shown to cause structural, functional, or biochemical alterations in cells to facilitate viral replication, virion assembly, and egress. In particular, the endomembranes can undergo dramatic changes because of the strategy of the viruses to hijack cell structures [11,12,13]. hRSV replication and transcription process occur in the cytoplasm and Golgi, but not in the nucleus [14,15,16]. Nonetheless, some viral proteins are imported into the nucleus. For example, the M2-1 viral protein promotes nuclei NF-kB activation and Rel-A [17], and the M protein interacts with zinc finger protein 2, SMAD3, and chromatin, inhibiting the host cell transcription [18,19]. NS1 and NS2 inhibit IRF-3, inhibit INF activation, and induce STAT2 degradation [20,21]. Furthermore, it has been reported that hRSV antagonizes the innate immune response mediated by MDA5 and MAVS [22,23]. Both proteins are part of the mitochondrial antiviral signaling system. In fact, hRSV interacts with the mitochondrial interface. This interaction impaired mitochondrial respiration by causing loss of mitochondrial membrane potential and increasing mitochondrial ROS [24]. The ultrastructural changes of the mitochondria, as a response to a viral infection, is probably one of the most studied examples of virus-induced organelle remodeling [25]. Yet we still do not fully understand how mitochondrial alterations contribute to viral replication and pathogenesis. Some viruses disrupt the mechanisms of mitochondrial dynamics to inhibit antiviral signaling. For example, hepatitis C virus enhances mitochondrial fission through the activation of Drp1, which is a protein associated with fission. Also, it inhibits innate immune responses and apoptosis by enhancing mitophagy [26]. Hepatitis B virus also inhibits the innate immune response and apoptosis by enhancing mitochondrial fragmentation and mitophagy through phosphorylation of Drp1 and upregulation of mitophagy-associated proteins Parkin and PINK1, which are ubiquitin kinases related with mitochondrial health [27]. In contrast, the measles virus (MeV) induces mitophagy to reduce the mitochondrial antiviral signaling protein levels and block the innate immune response [28].

The effect of hRSV infection on the mitochondria is not well understood. However, proteomic studies on hRSV-infected cells have identified changes in protein levels of several nuclear-encoded mitochondrial proteins such as voltage-gated anion channel (VDAC) protein and prohibitin (PHB). These proteins play a critical role regulating mitochondrial structure, function, and biogenesis [29,30]. Furthermore, it was found that hRSV infection hijacks the mitochondria during replication by redistributing them to the perinuclear region. This spatial redistribution is a microtubule/dynein-dependent process that is induced by the viral infection and results in changes in mitochondrial membrane polarization. These changes decrease the mitochondrial membrane potential and increase the levels of reactive oxygen species levels [31]. Nonetheless, little is known about hRSV-induced alterations in mitochondrial dynamics.

In this study, we explored the ultrastructural and functional changes of the mitochondria of HEp-2 cells infected with hRSV-A ON1. We found that hRSV infection induces alterations in mitochondria size (area) and Feret’s diameter. Mitochondria of infected cells have an aberrant morphology: there is damage to the cristae, and there is a loss in the membrane integrity. We found that during early infection time points, hRSV infection promotes mitochondrial fusion associated with mitofusins (Mfn1) mRNA downregulation. Furthermore, except for transient overexpression at 24 h.p.i., we also found down-regulation of VDAC2. Surprisingly, the mRNA for the kinase-mediated mitophagy (PTEN-induced serine/threonine kinase, PINK1) mRNA was also suppressed during infection. Our results suggest that hRSV infection induces changes in mitochondrial dynamics that regulate activities linked to the fusion processes and the antiviral signaling pathways in mitochondria and inhibits mitophagy. Overall, these changes in mitochondrial dynamics might imply a mechanism that contributes to the pathogenesis of RSV that could be essential to allow the formation of syncytia.

## 2. Materials and Methods

### 2.1. Cell Line Culture, Viral Stock and Treatments

HEp-2 cells (ATCC, CCL-23) were grown, following previous protocols for hRSV infections, in 6-well culture plates in 5% CO_2_ and 37 °C in DMEM (ThermoFisher, Grand Island, NY, USA) supplemented with 5% fetal bovine serum (Gibco, Grand Island, NY, USA) and 1% streptomycin (Pisa, Guadalajara, Mexico) and penicillin (PiSA, Guadalajara, Mexico) [32,33,34,35,36,37].

The viral stock was generated from a clinical sample positive for hRSV, the genotype was assigned by sequencing the ectodomain region of the G gene and the isolated was used subsequently to infect confluent monolayer HEp-2 cells. The presence of the cytopathic effect of the virus was identified (by light microscopy after five days, and the presence of the virus was confirmed by qRT-PCR (with absolute quantification of the number of viral copies); a subculture was performed at MOI 5, and virus concentration was carried out at low speed starting from 25.2 x10^6^ infected cells at MOI 5 based on the protocol of Rayaprolu et al. [38]. The viral stock was quantified by qRT-PCR using the primers in Table 1 and stored for less than six months refrigerated at 4 °C.

Upon reaching 95% confluence, culture media was replaced by 1x PBS, and the cells were inoculated with a viral stock of hRSV NA1-ON1 genotype at a multiplicity of infection (MOI) of 10 and incubated for two hours with constant agitation every 15 min. After the absorption time, the inoculum was removed, and culture media supplemented with 1% fetal bovine serum was added and cultures were incubated at 37 °C with 5% CO_2_. Infected cells were incubated for 2, 24, 48, 72, and 96 h; afterward, cells were washed with 1x PBS and fixed with 2% glutaraldehyde and 0.05% picric acid in 0.1-M cacodylate buffer pH 7.4 for 1 h at room temperature and then stored at 4 °C until processing for transmission electron microscopy (TEM). Cultures used for RT-qPCR were processed under the same conditions up until harvesting; at that point, they were incubated for 10 min with 500 μL of trypsin (TrypLE Express, Grand Island, NY, USA), and 1 mL of supernatant was taken from cultures corresponding to each condition to detach the cells. The sample was homogenized and placed and maintained at 4 °C until RNA extraction.

### 2.2. Transmission Electron Microscopy

Thin-section TEM was used to characterize the ultrastructural modifications of infected cells. Cultures were processed using the previously described methodology [39]. Briefly, after the samples were fixed as described in the previous section, these were washed three times with 0.1 M sodium cacodylate buffer pH 7.2; the samples were further fixed with 1% osmium tetroxide in 0.1 M sodium cacodylate for one hour at room temperature in the dark. The cultures were rinsed twice with the sodium cacodylate buffer and once with 0.1 sodium acetate buffer pH 4.5, block-stained with 0.5% uranyl acetate in 0.1 M sodium acetate for one hour at room temperature in the dark. Subsequently, the cells were washed three times with the 0.1 M sodium acetate and two times with MilliQ water before being left to incubate in MilliQ water overnight. The following day, cells were dehydrated with ethanol with increasing concentration (35%, 50%, 70%, 95%, and 100%) for 10 min and three times for each concentration, then cells were rinsed three times with Embed 812 resin (Electron Microscopy Science, Hatfield, PA, USA). Finally, the cells were embedded in the EMBed 812 resin and polymerized for 48 h at 55 °C. Ultrathin sections (70 nm) were obtained using an EM-UC7 ultramicrotome (Leica Inc., Karnataka, India) and transferred to 100-mesh copper grids and stained with 0.5% uranyl acetate and 0.5% lead citrate. After staining, the grids were kept in closed Petri dishes containing dry pellets of NaOH and silica. The samples were visualized in a JEM-JEOL-2100 transmission electron microscope at 200 kV using a Gatan 4K camera. The digital electron micrographs were recorded with the Digital Micrograph (Gatan Inc., Pleasanton, CA, USA) software and analyzed with ImageJ and Fiji (NIH, Bethesda, ML, USA). Each sample was done in duplicate.

### 2.3. Qualitative Morphological Evaluation

Micrographs were evaluated qualitatively using a method to correlate the degree of ultrastructural damage with its specified stage of mitochondrial injury using the previously described methodology [40,41]. This classification has four classes and are described in Table 2.

### 2.4. Morphometric Analysis

The morphometric analysis was performed using the Fiji software version 2.9.0/1.53 (NIH) by randomly selecting 25 micrographs per sample. The mitochondrial shape descriptors and size measurements were obtained by manually selecting the contours. The size descriptors studied were the surface area (or mitochondrial size); the perimeter, and the Feret’s diameter (which is the largest distance between two points). The shape descriptors analyzed were the aspect ratio (AR) (major axis/minor axis) which represents the “length-to-width ratio”; the form factor (FF) (perimeter^2^/4π × Surface area) that reflects the complexity and branching appearance of mitochondria; the circularity (4π × surface area/perimeter^2^), and the roundness (4*surface area/π × major axis^2^). These last two parameters are two-dimensional indices of sphericity with values of 1 denoting perfect spheroids. Once the data was obtained, the calculated values were imported into Microsoft Excel and Prism 9 (GraphPad Software, San Diego, CA, USA) for data analysis.

### 2.5. RNA Extraction

Viral RNA was extracted from the cell cultures at different times post-infection (2-, 24, 48-, 72- and 96-h post-infection) using the QIAamp^®^ Viral RNA extraction kit (Qiagen Brand 52906, Germantown, MD, USA) and was eluted in the AVE buffer (RNAse-free water containing 0.04% sodium azide). Total RNA concentration and purity was determined using a spectrophotometer nanodrop. Only samples with an OD_260nm_/OD_280nm_ between 1.8 and 2.0 were further processed. Samples were stored at −80 °C until use.

### 2.6. Analysis of Gene Expression by Reverse Transcription Polymerase Chain Reaction (RT-qPCR)

Quantitative RT-PCR was performed using the Luna^®^ Universal One-Step RT-qPCR kit (New England BioLabs, Ipswich, MA, USA) in a Mic qPCR Cycler (Bio Molecular Systems) magnetic induction thermocycler, using the glyceraldehyde-3-phosphate dehydrogenase (GAPDH) gene as an endogenous gene. Primers used to quantify expressions of mRNAs related to the mitophagy process PINK1 (NM_032409.3), VDAC2 (NM_003375.5), and Mfn1 (NM_033540.3) mRNA expression are shown in Table 2. The reaction (10 μL total volume) contained 5 μL Luna Universal One-Step Reaction Mix (2×) (New England BioLabs), 2 μL RNA (10 ng/mL), 2.1 μL nuclease-free water, 0.4 μL of forward and reverse primers each at a concentration of 10 μmol/L, and 0.5 μL of Luna WarmStart^®^ RT Enzyme Mix (20×) (New England BioLabs, Ipswich, MA, USA). The reaction process consisted of a retrotranscription of 55 °C for 10 min and 95 °C for 1 s, followed by 40 cycles of 95 °C for 10 s and 60 °C for 30 s, and finally, a dissociation curve from 60 °C to 95 °C with a 0.3% increase. Subsequently, the 2ΔΔCt between the target gene and the housekeeping GAPDH gene was calculated using the previously established method [42]. GAPDH was chose as a housekeeping gene based on previous studies with hRSV [43,44]. Furthermore, the relative expression of this housekeeping gene in the infected cells with respect to the mock-infected cells was not statistically significantly different at any time point (Appendix A). All experiments were performed in duplicate, and we analyzed two biological replicas.

### 2.7. hRSV Viral Copy Quantification by Reverse Transcription Polymerase Chain Reaction (RT-qPCR)

10 ng of RNA were used for the Retrotranscription using the RevertAid H kit Minus First Strand cDNA Synthesis Kit (Thermo Scientific, Waltham, MA, USA). First Strand cDNA Synthesis Kit (Thermo Scientific, Waltham, MA, USA). From this cDNA, PCR was performed for absolute quantification of RSV viral copies using the following primers: MCVF (5′-GGCAAATATGGAAACATACGTGAA-3′) and MCVR (5′-TCTTTTTCTAGGACATTGTAYTGAACA-3′), at a final concentration of 0.25 μm, which are directed to the intergenic region between the RSV P and M genes and generate an 87 bp product. As a control for the standard curve, a 2752 bp pUCIDT that has a 500 bp synthetic construct including the quantification PCR target site was used. qPCR was performed with the Maxima SYBR Green/ROX qPCR Master Mix (2x) kit (Thermo Scientific, Waltham, MA, USA) as indicated in the insert, the cDNAs obtained were analyzed in duplicate, as well as the 6 points of the standard curve. The data of the experiment were considered adequate if the standard curve correlation coefficient was greater than 0.98. The quantification was carried out by means of the linear regression method using the parameters of the line from the standard curve. All experiments were performed in duplicate, and we analyzed two biological replicas.

### 2.8. Statistical Analysis

The statistical analysis was performed using GraphPad Prism 9 (GraphPad, San Diego, CA, USA). The data distribution for each of each data set was assessed using a Shapiro-Wilk as a test of normality to assess whether the data series conformed to a normal distribution. Nonparametric data were described as median (interquartile range). Comparisons within groups were made using the Friedman test. Comparisons between groups were made using the Mann–Whitney U test. For categorical variables we performed a 2 x 2 analysis and the Fisher’s exact test. In all cases, *p* < 0.05 was considered statistically significant. The data in Figures that plot the Feret’s diameter, area, and RT-qPCR, are shown as boxplot and the lines indicate the data between the lower quartile (25%) and the lower extreme, and between the upper quartile (75%) and the upper extreme.

## 3. Results

### 3.1. Effects of hRSV ON1 Genotype Infection on the Morphology of the Mitochondria of the HEp-2 Cell Line

The mitochondrial morphology of the control HEp-2 (mock-infected) and infected cells were characterized at the ultrastructural level by thin-section TEM. The morphology of the mitochondria in control HEp-2 cells remained consistent across all time points, with the classical shape, a double mitochondrial membrane, and defined cristae (Figure 1A–D). In contrast, we found significant changes in the mitochondrial ultrastructure of hRSV ON1-infected cells. At 2-h post-infection (hpi) (Figure 1E), irregular or swirling mitochondrial cristae were seen, and at 24-hpi (Figure 1F), the mitochondria have swollen matrix, discontinuous outer membranes, and fragmented cristae. At 48-hpi (Figure 1G), the mitochondria had varied shapes and sizes, as well as a discontinuous outer membrane and fragmented cristae. Finally, at 72-hpi (Figure 1H), the mitochondria have irregular or swirling cristae.

### 3.2. Morphological Characterization of Mitochondria Infected with RSV ON1 in the HEp-2 Cell Line

This organelle was classified into four classes to determine the degree of mitochondrial damage in HEp-2 cells. Class I (Figure 2A) corresponds to normal mitochondria with densely packed longitudinal cristae and an electron-dense matrix. Class II (Figure 2B) is characterized by bulging, irregular, or swirling cristae that have lost their orientation and/or narrowness. Class III (Figure 2C) presents varied shapes and sizes, with discontinuous mitochondrial membranes, fragmented cristae, swollen, and electro-lucent matrix. Finally, class IV (Figure 2D) refers to mitochondria with disrupted membranes, disrupted mitochondrial cristae, and diffuse appearance. Between 50 and 150 mitochondria were evaluated in each sample, and the degree of mitochondrial damage was observed to increase gradually over time.

We analyzed the ultrastructural changes at 2-hpi in the control group and 98.3% of the mitochondria exhibited normal morphology (Figure 2E) and only 1.7% exhibited an alteration in their morphology, particularly damage in the mitochondrial cristae (Figure 2F). In contrast, in the hRSV infected group, 52% of the mitochondria exhibited normal morphology (Figure 2E), 34.7% exhibited damage in the mitochondrial cristae (Figure 2F), and 13.3% showed damage to cristae and mitochondrial membrane (Figure 2G).

At 24-hpi, in the control group, 59.3% of the mitochondria exhibited normal morphology (Figure 2E), 20.3% exhibited damage in the mitochondrial cristae (Figure 2F), 17% showed damage to cristae and mitochondrial membrane (Figure 2G), and 3.4% of the mitochondria showed a disrupted morphology (Figure 2H). In the infected groups, 6.6% of the mitochondria exhibited normal morphology (Figure 2E), 50% exhibited damage to the mitochondrial cristae (Figure 2F), 38.5% showed damage to cristae and mitochondrial membrane (Figure 2G), and 4.9% exhibited a destroyed morphology (Figure 2H).

The control group at 48-hpi showed that 83.6% of the mitochondria exhibited normal morphology (Figure 2E), 7.3% exhibited damage to the mitochondria cristae (Figure 2F), and 9.1% showed damage to cristae and mitochondrial membrane (Figure 2G). In the infected cells, 2.4% of the mitochondria exhibited normal morphology (Figure 2E), 23.5% exhibited damage to the mitochondrial cristae (Figure 2F), 62.4% showed damage to cristae and mitochondrial membrane (Figure 2G), and 11.7% exhibited a destroyed morphology (Figure 2H).

At 72-hpi, in the control group, 32.7% of mitochondria had normal morphology (Figure 2E), 64.5% had damaged mitochondrial cristae (Figure 2F), and 2.8% exhibited damage in the cristae and the mitochondrial membrane (Figure 2G). In the infected cells, 7.7% of the mitochondria exhibited normal morphology (Figure 2E), 52.6% exhibited damage to the mitochondrial cristae (Figure 2F), 38.5% showed damage to cristae and mitochondrial membrane (Figure 2G), and 1.2% showed morphological disruption (Figure 2H).

Finally, when the control group was analyzed at 96-hpi, 67.4% of the mitochondria had normal morphology (Figure 2E), 24.7% has damaged mitochondrial cristae (Figure 2F), and 7.9% showed damage to cristae and mitochondrial membrane (Figure 2G). In contrast, in the infected cells, 17.2% of the mitochondria exhibited normal morphology (Figure 2E), 50.3% exhibited damage to the mitochondrial cristae (Figure 2F), 23.6% showed damage to cristae and mitochondrial membrane (Figure 2G), and 8.9% exhibited a destroyed morphology (Figure 2H).

In the control group Fisher’s exact test indicates that the prevalence of mitochondria in the mock infected cells with normal morphology (Class I) is higher compared to abnormal morphology of mitochondria at any time point (Classes II, III, and IV) (*p* < 0.0001). Furthermore, the same test indicated that the infected cells have a higher number of mitochondria with abnormal morphology (Classes II, III, and IV) than the mock infected cells (*p* < 0.0001).

These results show that the hRSV-ON1 infection alters the structure of the mitochondria; specifically, the morphology of mitochondrial cristae and the integrity of mitochondrial membranes are greatly affected by the infection. These results suggest that hRSV-ON1 infection is related to alterations in the function of mitochondrion membrane-related proteins and pathways.

### 3.3. Morphometric Analysis of RSV ON1-Infected Mitochondria in the HEp-2 Cells Line

Subsequently, morphometric measurements were performed in HEp-2 cells infected cells and compared to the mock-infected cells (Figure 3). The mitochondria surface area was used as a size descriptor, and 57 mitochondria from each time point were measured. The area of the control samples was examined over time. When comparing this shape descriptor as a function of time we only observed statistically significant differences at 2- and 48 h.p.i. (Figure 3A) (U-Mann Whitney). At 2-h.p.i. the mean mitochondria area was 0.18 ± 0.11 (±SD) μm^2^, at 24-h.p.i. the mean area to 0.57 ± 0.31 μm^2^. At two subsequent time points, the mean area did not change substantially: at 48- and 72-h.p.i the mean areas are 0.54 ± 0.32 μm^2^ and 0.53 ± 0.30 μm^2^, respectively. Finally, at 96-h.p.i. the mean area decreased to 0.41 ± 0.24 μm^2^ (Figure 3B). Using a Friedman test, we found a statistically significant difference between the area at 2-h.p.i. and the areas at each of the times of infection. In addition, we found a statistically significant difference between 24-h.p.i. and 96-h.p.i. when the area of the mitochondria decreases.

The mean mitochondrial area of the infected cells between the 2- and 48-h.p.i 0.25 ± 0.18 μm^2^, 0.52 ± 0.42 μm^2^, and 0.71 ± 0.48 μm^2^. The area decreased at 72- and 96-h.p.i. the area was 0.57 ± 0.21 μm^2^ and at 96-h.p.i. the area was 0.43 ± 0.23 μm^2^ (Figure 3C). Using the Friedman test, we found statistically significant differences between the mean area at 2-h.p.i. and the mean areas at each time of infection. In addition, we found a statistically significant difference between 48-h.p.i. and 96-h.p.i. when the area decreases in this last time point. These results indicate that in infected cells, the increase in area is sustained until 48-h.p.i. and after this, the mean area decreases as a function of time approaching that of the uninfected cells. We did not find statistical differences when comparing the mitochondrial area between both groups at most time points except for 2- and 48-h.p.i.

Another size descriptor, Feret’s diameter, was used to confirm these findings. As with the surface area, there were statistically significant changes over time. In the mock-infected samples at 2-h.p.i. mitochondria Feret’s diameter was 0.66 ± 0.29 (mean ± SD) μm, at 24-h.p.i. the Feret’s diameter increased to 1.1 ± 0.44 μm. However, at the following two time points, the diameter did not change substantially (1.1 ± 0.53 μm and 1.0 ± 0.42 μm, at 48- and 72-h.p.i). Finally, at 96-h.p.i. the diameter deceased to 0.89 ± 0.35 μm (Figure 4A). Using the Friedman test, we found a statistically significant difference between the size at 2-h.p.i. and the sizes at each of the time points. In addition, we found a statistically significant difference between the 24- and 96-h.p.i. when the size decreases in this last time. These results indicate that in the control group, there was an increase in size from 24-h.p.i., the diameter was maintained over time and subsequently decreased in the last time of infection.

Figure 4A shows the differences in the Feret’s diameter between groups at each time point and as in Figure 3A at 2 and 48 h.p.i. the differences between this shape descriptor are statistically significant (U-Man Whitney). In the infected cells, the Feret’s diameter at 2-h.p.i. was 0.78 ± 0.27 (mean ± SD) μm, and at 24- and 48-h.p.i. the diameter increased to 1.1 ± 0.66 μm and 1.3 ± 0.72 μm, respectively. Finally, the diameter decreased at 72- and 96-h.p.i. from 1.1 ± 0.35 μm to 0.92 ± 0.36 μm, respectively (Figure 4B). Using the Friedman test, we found that there is a statistically significant difference between the size at 2-h.p.i. and the areas at each of the post infection times. In addition, we found that there is a statistically significant difference between the 48- and 96-h.p.i. when the size decreases in this last time. These findings suggest that hRSV ON1 infection is associated with an increase in mitochondrial size in Hep-2 cells that is different from the one observed in the control cells.

Finally, the complexity of mitochondria was evaluated by correlating two morphometric descriptors, the aspect ratio (AR) and the form factor (FF). In the first place, we compared the shape descriptors at 2-h.p.i. and found that the observed shapes of the mitochondria (i.e., round and elongated) in control and infected cells were not significantly different from one another (AR = 2.0 ± 0.88 vs. 2.3 ± 1.0 for infected cells; FF = 1.3 ± 0.31 vs. 1.4 ± 0.35 for infected cells). We found no statistically significant differences when the equations of the linear regressions for the correlations were compared using the Student *t*-test for both groups. A value of *p* > 0.05 was obtained (Figure 5A).

However, at 24-h.p.i. (Figure 5B) we found that the correlations between AR and FF for the control and infected cells were significantly different from each other (AR = 1.7 ± 0.68 vs. 1.8 ± 0.68 for infected cells; FF = 1.2 ± 0.26 vs. 1.3 ± 0.44 for infected cells). Statistically significant differences were obtained when comparing the equations for the linear regression of both groups (*p* < 0.001). Figure 5B shows that in the infected groups at 24-h.pi. the data points can be separated into two populations; one group of data points overlaps with all the data points of the mock-infected cells, while another group deviates considerably from this. This suggests that at this time point, there are two populations of mitochondria in the infected cells and one of them is more elongated than in the control cells.

At 48 h-post-infection (Figure 5C) there is a similar effect as 24-h.p.i. However, the differences between the infected and mock-infected samples are less pronounced than at the previous time point. The mean values for AR and FF observed that in both control and infected cells were significantly different in both groups (AR = 1.8 ± 1.1 vs. 1.8 ± 0.93 infected cells; FF = 1.2 ± 0.39 vs. 1.3 ± 0.42 for infected cells). We also found that in the infected cells, there is a population that is more elongated than in the controls. This can be seen by the data points that do not overlap with the control and by the fact that there is a statistically significant difference (*p* < 0.001) between the correlations for both samples. At 72- (Figure 5D) and 96-h.p.i. (Figure 5E) there are no statistically significant differences found when comparing the regression of the correlations. Overall, these findings indicate that RSV ON1 infection causes an increase in the size and elongation of mitochondria in HEp-2 cells at 24 and 48-h.p.i.

### 3.4. Effects of RSV ON1 Infection on HEp-2 Cell Line Mitochondrial Function Genes Expression

In order to determine if the changes in the mitochondria morphology are related to functional changes, we investigated if the infection with hRSV-ON1 is associated with alterations in homeostasis and mitochondrial function. Three mRNAs associated with mitochondrial function were analyzed: Mfn1, VDAC2, and PINK1. The Mfn1 gene encodes mitofusin 1, a protein related to mitochondrial outer membrane fusion. The VDAC2 gene encodes the voltage-dependent anion-selective channel 2, whose functions are related to oxidative metabolism and apoptosis. The PINK1 gene encode protein related to the process of mitophagy. The relative expression of these genes was normalized with GADPH, and the Friedman test was performed for each group over time. Infection with hRSV ON1 resulted in downregulation of the Mfn1 (Figure 6), VDAC2 (Figure 7), and PINK1 (Figure 8) mRNAs. This expression was found to have a maximum in the infected group at 24-h.p.i. However, this expression is dramatically reduced at later time points. In the control group, the relative expression of these genes was constant.

In the case of the expression of PINK1 mRNA (Figure 8), there are no significant differences in the mock infected cells at different time points. However, in hRSV infected cells the expression of this mRNA is downregulated, specially at 48-, 60-, 72- and 96 h.p.i.

These data indicate that the expression of genes related to mitochondria elongation and apoptosis are down-regulated by hRSV infection, except during early stages of infection when transient overexpression of VDAC2 was observed. As such, it appears that hRSV-ON1 infection does not induce mitophagy.

## 4. Discussion

It has been previously documented that hRSV causes changes in the host cell’s mitochondria that promote viral replication. In particular, previous studies show that mitochondria accumulate perinuclearly during early infection [24], while at the late infection stages the distribution near the microtubule organizing center is asymmetric [31]. Furthermore, hRSV infection has been shown to inhibit mitochondrial respiration, disrupt membrane potential, and increase reactive oxygen species production to promote viral replication [45]. These results demonstrate the importance of mitochondria during infection. However, few studies have examined changes in mitochondrial dynamics, including changes that affect the size, shape, and function of this organelle. To study the changes in mitochondrial dynamics, we used thin-section TEM and RT-qPCR to examine changes in mitochondrial structure and function induced by hRSV ON1 in Hep-2 cells.

We observed that the percentage of class 2 mitochondria in the mock-infected cells increased at 24- and 72-h.p.i. It is important to point out that at time 0-h.p.i the cells were incubated with either virus (in PBS) or PBS, and after two hours, the inoculum was replaced with complete DMEM. Also, the media was replaced after 48-h.p.i. Therefore, it is likely that the replacement of PBS with complete MDEM and/or the media replacement process could stress the cells, resulting in an increase in the percentage of Class 2 mitochondria at 24 and 72 h.p.i. Nonetheless, we found that, independently from these changes, hRSV ON1 induced changes in mitochondrial morphology by inducing outer membrane discontinuities, altering the shape of mitochondrial cristae, and inducing the formation of swirling or fragmented cristae. In contrast, the mitochondrial morphology of hRSV-uninfected cells did not change over time (mitochondria had clearly defined cristae and membranes).

While most studies with hRSV use MOIs between 0.1 and 3.0 [24,29,30,31,45,46] we decided to use an MOI of 10 to be able to detect ultrastructural changes, which is one of the limitations of this study. In previous studies with Zika (ZIKV) [47] and Chikungunya (CHIKV) [48] virus, we used low MOIs for thin-section TEM. However, preliminary experiments with hRSV at MOIs of 0.1 and 1.0 showed that detecting ultrastructural changes in the mitochondrion was extremely challenging at these MOIs (data not shown). Furthermore, in the cases of CHIKV [48] and ZIKV [47] infections at an MOI of 1.0 lead to complete destruction of the cellular monolayer within the first 72 h.p.i.; however, with hRSV, infection at an MOI of 10 does not result in complete disruption of the monolayer even at 96-h.p.i. and hence, we sought to use a high MOI to increase the chance of finding ultrastructural changes. Nonetheless, our results consistent with published studies. In a previous study, hRSV was inoculated into guinea pig middle ears, and the infection generated swollen mitochondria in epithelial and mononuclear cells [49]. Another study showed mitochondrial damage in ciliated cells infected with hRSV [46]. Additionally, mitochondrial alterations have also been reported with other viruses, such as HCV [27] and SARS-CoV-2 [50]. Our findings suggest that hRSV ON1 alters mitochondrial morphology, especially the shape of mitochondrial membranes and cristae, during infection.

We then further examined the changes in mitochondrial morphology and quantified several shape descriptors. By evaluating the surface area, Feret’s diameter, and complexity, we observed an increase in mitochondrial size due to elongation at early times. Based on our observations, we determined that hRSV-ON1 induces the formation of elongated mitochondria. On the hand, the frequency of damaged mitochondria increases with time and at 24- and 48-h.p.i. the morphology of the mitochondria in the infected cells is different compared to the mock-infected cells. On the other hand, the surface area and Feret’s diameter are similar between infected and mock-infected cells. These observations suggest that, most likely, the class 3 and 4 mitochondria are degraded over time, and thus, in the later time points, we only detected class 1 and 2 mitochondria. This biases our observations toward healthy cells. Nonetheless, this could also indicate that hRSV-ON1 modulates mitochondrial dynamics to inhibit apoptosis and increase the chance of generating syncytia.

Previously, a study using confocal microscopy reported mitochondrial elongation in Hep-2 cells upon hRSV infection [51]. Elongation effects were also observed with MeV [51]. Furthermore, SARS-CoV promotes mitochondrial elongation in HEK293T and THP-1 cells through the expression of the viral accessory protein ORF-9b [52]. It has also been shown that dengue virus infection in hepatoma cells leads to mitochondrial elongation by decreasing phosphorylation of dynamin-related protein 1 (DRP1) [53]. Recently, H1N1 influenza virus was reported to induce a state of hyperelongated mitochondria in A549 cells; this effect was promoted by re-localization of the DRP1 protein [54]. Our results suggest that hRSV ON1 induces the formation of hyperfused mitochondria at early stages. However, the mechanism by which this induction occurs needs further investigation.

Given the fact that our morphometric analysis could be biased toward healthy cells or organelles (i.e., those that are too damaged can be hard to find), we sought to study the expression of mRNAs related to the function of the mitochondria. First, we investigated Mfn1, which mediates mitochondrial outer membrane fusion through guanosine-5 hydrolysis-driven homo- and heterotypic interactions with triphosphate (GTP) [55]. This protein is essential for mitochondrial quality control and homeostasis, as well as for the regulation of numerous biological processes, including the immune response [56]. During viral infection, the immune response begins when viral RNA is detected in the cytosol by RIG-I-like receptors (RLRs), specifically RIG-1, which have two amino-terminal caspase activation and recruitment domains (CARDs); these domains interact with mitochondrial antiviral signaling protein, which are proteins of the mitochondrial outer membrane, and its activation is essential for initiating the production of type I interferons (IFN) and pro-inflammatory cytokines [57]. It has been reported that Mfn1 is required for efficient RLR signaling. When the RLRs are activated, Mfn1 promotes mitochondrial fusion/elongation. Conversely, when fusion is inhibited by Mfn1 silencing, activation of type I interferons (IFN) and pro-inflammatory cytokines decreases [58,59].

Our results showed that hRSV ON1 infection induces down-regulation of Mfn1 expression. These results might explain, in part, the shape and size descriptors described above. However, there is no direct evidence that mitochondrial elongation is related to Mfn1. In one study, Sirtuin 1 (SIRT1) was reported to regulate mitochondrial function in hRSV-infected dendritic cell [60]. The relationship between SIRT1 and Mfn1 has been previously studied and showed that mitochondrial elongation is regulated by SIRT1-mediated stabilization of Mfn1 [61]. On the one hand, this could indicate the role of Mfn1 as a modulator of the immune response to viral infections. However, more studies are needed to confirm this association. On the other hand, in other viruses, such as dengue virus [62] and measles virus [51], the relationship between hyperfusion and Mfn1 expression during the infection processes has been directly studied.

The expression of mRNA for VDAC, the most abundant mitochondrial membrane protein, was also evaluated in its isoform 2 (VDAC2). This protein is involved in a wide range of processes, such as the passage of ATP outside the mitochondria, the release of anions superoxide, as well as in the apoptosis process [63]. One of the roles of the VDAC2 protein is to regulate the apoptosis process through the release of pro-apoptotic factors such as cytochrome C. It has been described that increased membrane permeability induced by VDAC2 overexpression could mediate cytochrome C release. This leads to the activation of caspases, proteolytic enzymes which are key proteins in the apoptotic process [64]. The relevance of this protein and its manipulation could have significant implications during viral infection. The quantification of VDAC2 mRNA showed a transient upregulation in early stages of infection, while notable downregulation occurred at later stages of infection. This behavior is similar to the one we found for Mfn1.

Previous studies have described that the hRSV M protein interacts with the VDAC protein. However, the exact molecular mechanism of this interaction and its effect on RSV replication are still unknown [29]. Furthermore, a previous study analyzing the mitochondrial proteome during hRSV infection in A549 cells found that the levels of different isoforms of the VDAC were significantly increased during the late stages of infection [30]. As another example, influenza virus PB1-F2 protein induces cell death through VDAC1. Although the induction of apoptosis by influenza virus may appear to be contradictory for efficient viral production, it is critical for influenza viral replication [65]. Our results are inconsistent with these previous studies, as we observed downregulation in late times of the infection. Apoptosis-related signaling pathways are inhibited by hRSV infection, which may be its survival strategy. More research is needed to understand the relationship between all of our observations.

PINK1 is a key mediator of mitochondrial quality control processes and activates the mitophagy process [66]. Unlike the expression of the previously described genes (Mfn1 and VDAC2), PINK1 down-regulation was less notable. PINK1 expression downregulation has been described previously. hRSV infection in mice primary peritoneal macrophages induces PINK1 downregulation. This same phenomenon was also observed when studying infection with vesicular stomatitis virus and herpes simplex virus [67]. These results indicated that PINK1 mRNA was downregulated following hRSV ON1 infection. This may indicate that hRSV ON1 infection inhibits the PINK1/Parkin-dependent mitophagy pathway. However, more research is needed to allow us to understand the mechanism of this inhibitory effect. To further understand this inhibitory effect, it is important to consider the dynamics of the PINK1/Parkin-dependent mitophagy pathway.

On the one hand, most of the ultrastructural changes were only noticeable in early times, before there was an exponential growth in viral RNA (see Appendix A). This could be due to the expression of early viral genes. At this stage it is unclear which genes are involved in these changes as there is a debate about whether hRSV gene expression follows a gradient or a steady-state regime [68,69]. Also, it seems to be a slight up-regulation of VDAC2 and Mnf1 only at early times, correlating almost perfectly with the alterations seen in Figure 5, before these genes are strongly down-regulated. On the other hand, at time points close and before the maximum viremia, most mitochondria have a normal morphology. This suggests that the downregulation of VDAC2, Mfn1, and PINK could be an indirect consequence of virus-induced cellular death, thus we were not able to observe abnormal mitochondria at these time points. It is important to point out that even at the peak of the viremia, we were not able to observe the complete destruction of the monolayer. Therefore, the lack of monolayer destruction and the abrupt drop in viremia at 72-h.p.i. suggest that the downregulation of VDAC2, Mfn1, and PINK might be an antiviral response that limits the progression of the infection and therefore at these time points most mitochondria have normal morphology.

## 5. Conclusions

This study provides evidence that hRSV-A ON1 infection induces significant changes in mitochondrial morphology and function in HEp-2 cells. The changes in mitochondrial size, cristae damage, and loss of membrane integrity indicate aberrant mitochondrial dynamics. The results show that hRSV infection promotes downregulation of several genes that are key for mitochondrial function. In addition, the study suggests that alterations in mitochondrial dynamics may impair antiviral signaling pathways and mitophagy and contribute to hRSV pathogenesis. These results shed light on the complex interplay between hRSV and mitochondria and offer potential targets for therapeutic interventions. The results of this study are promising as they can serve as a basis for the development of new antiviral therapies to combat hRSV. Further research is needed to determine how these findings can be used to improve the management of hRSV.

## Figures and Tables

**Figure 1 viruses-15-01518-f001:**
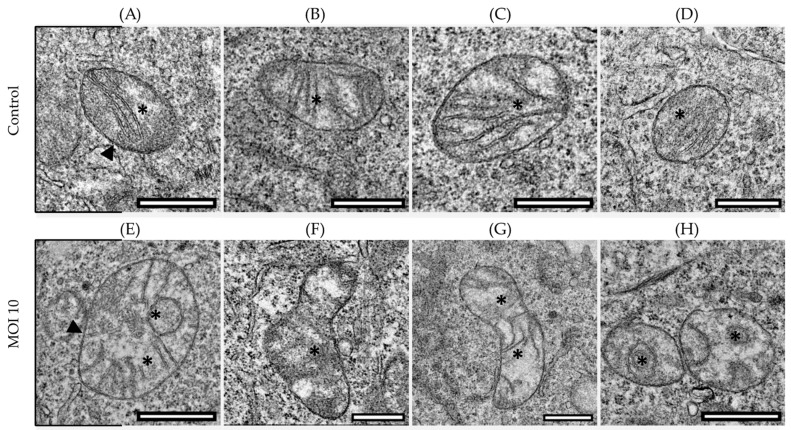
Representative micrographs of the ultrastructural characterization of mitochondria from hRSV ON1-infected HEp-2 cells. (**A**–**D**) Mock-infected samples at different time points (2-, 24-, 48-, 72-h.p.i. respectively). In these micrographs the mitochondrial morphology is maintained, the mitochondrial membrane is intact, maintaining the inner and outer membrane (arrowhead), and the mitochondrial cristae are transversely oriented (asterisks). On the contrary (**E**–**H**) show the infected cells at different time points (2-, 24-, 48-, 72-h.p.i. respectively) and there are different degrees of damage. (**E**) There are swirling cristae, which have lost their length and are no longer found transversally (asterisk). the mitochondrial membrane is intact (arrowhead). (**F**) The mitochondrial membrane is diffuse (arrow), a swollen matrix with shortened and diffuse cristae (asterisk). (**G**) There is damage to mitochondrial cristae, which have lost their length and are shortened. Finally, in (**H**) there are swirling cristae with shortened length (asterisks). The scale bar corresponds to 0.5 µm in the control and 0.75 µm in the infected.

**Figure 2 viruses-15-01518-f002:**
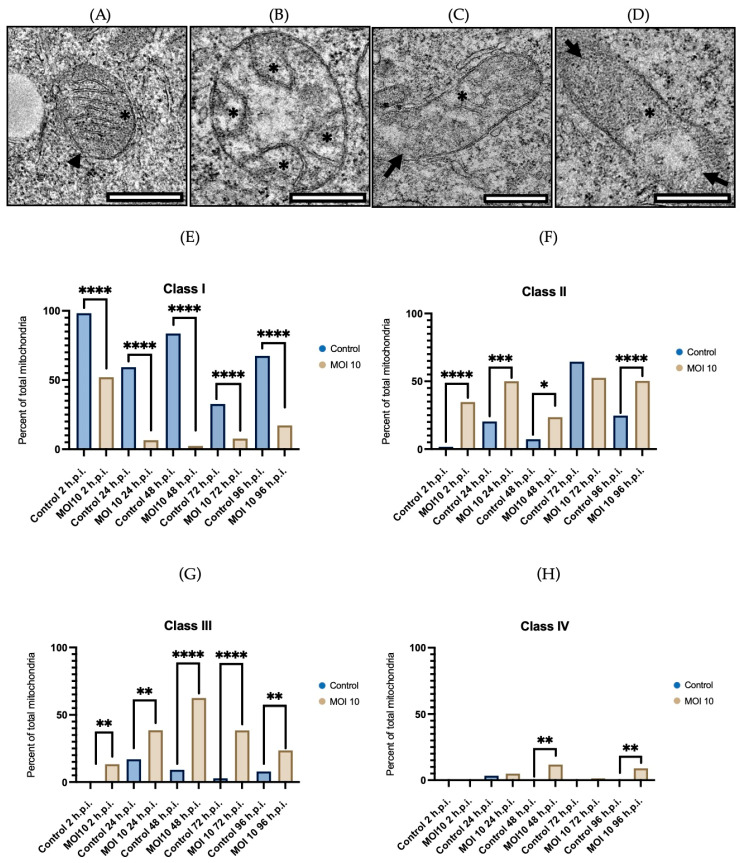
Representative micrographs illustrating the sorting of mitochondria from hRSV ON1-infected HEp-2 cells. Representative thin-section TEM micrographs according to the level of abnormality into four classes: class I (**A**) is characterized by having a defined membrane (arrowhead), in addition to transversely oriented cristae (asterisk); class II (**B**) is defined by a conserved membrane (arrowhead), but there is damage at the level of cristae that are shortened or swirled (asterisk); class III (**C**) corresponds to a mitochondrion with fragmented cristae (asterisk) and damaged membrane (arrow); and class IV (**D**) corresponds to the mitochondria with outer membrane broken and discontinuous (arrow) as well as deficient cristae (asterisk). (**E**) In the control cells, class I mitochondria predominates over time. (**F**,**G**) In contrast, in RSV ON1-infected cells there is a predominance of class II and class III mitochondria. (**H**) There are only a few mitochondria class IV both groups; however, there is a higher frequency of this class in the infected group at 48- and 96-h.p.i. Fisher’s exact test was performed for each class and the statistically significant differences are shown as an asterisk above the diagrams * corresponds to *p* ≤ 0.03, ** *p* ≤ 0.002, *** *p* ≤ 0.0002 and **** *p* ≤ 0.0001. Each sample was done in duplicate.

**Figure 3 viruses-15-01518-f003:**
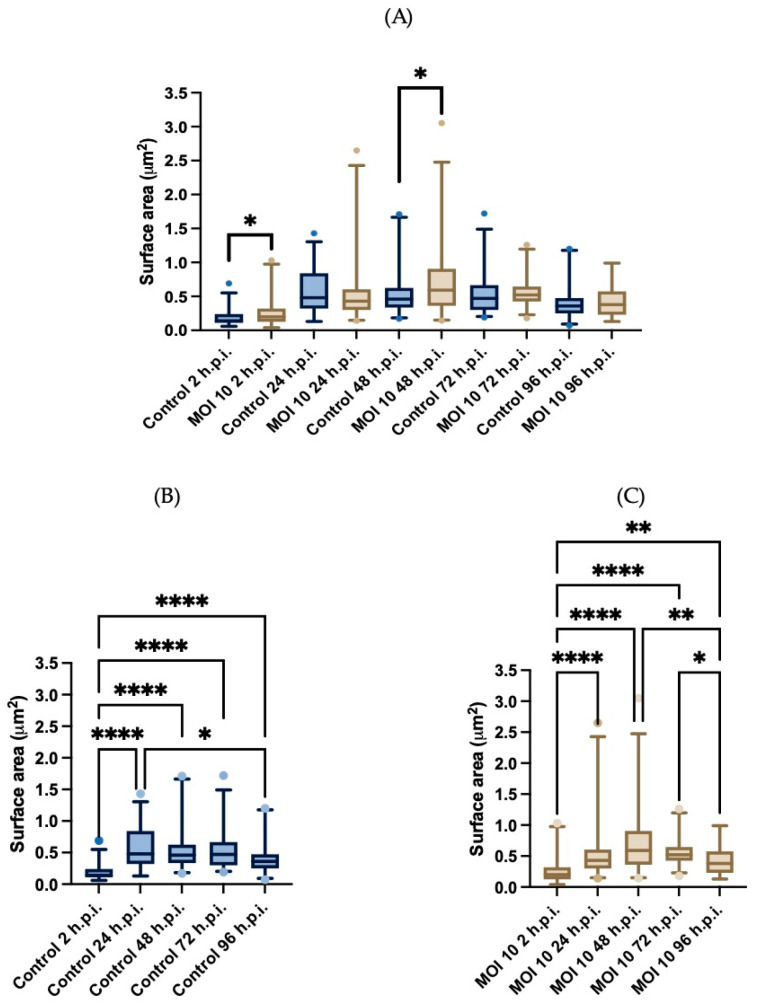
Morphometric analysis of the surface area of mitochondria represented in box plots. The surface area was compared between times points (**A**), and withing groups at different times post infections. (**B**) The group of the control HEp-2 cells (*n* = 57) is represented in a blue boxplot, and (**C**) the group of the HEp-2 cells infected with the hRSV ON1 MOI = 10 (*n* = 57) represented in a light brown color box plot. Data were analyzed using the Friedman test and statistically significant differences are shown in the form of an asterisk above the box plots, * corresponds to *p*
≤0.03, ** corresponds to *p*
≤0.002, and **** corresponds to *p*
≤0.0001. A U-Mann Whitney test was performed for the comparison between the control and infected group at each of the time post infection. The bars indicate the data between the lower quartile (25%) and the lower extreme and the upper quartiles (75%) and the upper extreme.

**Figure 4 viruses-15-01518-f004:**
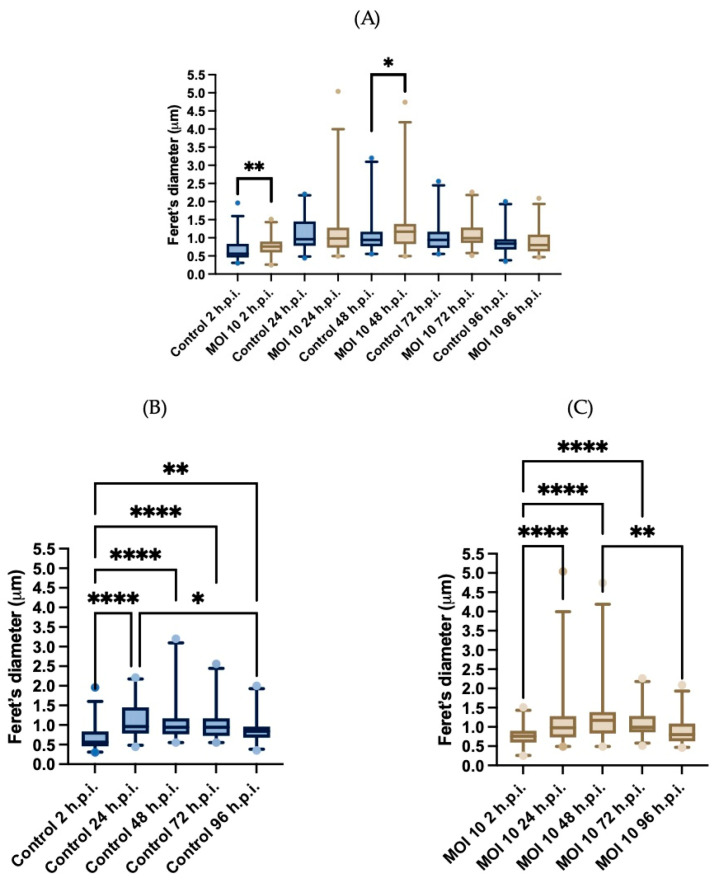
Morphometric analysis of the Feret’s diameter of mitochondria represented in box plots. (**A**) Feret’s diameter was compared between time points, and within groups at the different times post infection, the group of the control HEp-2 cells (*n* = 57) represented in a blue box plot (**B**) and the group of the HEp-2 cells infected with hRSV ON1 MOI10 (*n* = 57) represented in a light brown box plot (**C**). Data were analyzed using the Friedman test and statistically significant differences are shown in the form of an asterisk above the box plots * corresponds to *p* ≤ 0.03, ** corresponds to *p* ≤ 0.002, and **** corresponds to *p* ≤ 0.0001. A U-Mann Whitney test was performed for the comparison between the control and infected group at each of the time post infection. The bars indicate the data between the lower quartile (25%) and the lower extreme and the upper quartiles (75%) and the upper extreme.

**Figure 5 viruses-15-01518-f005:**
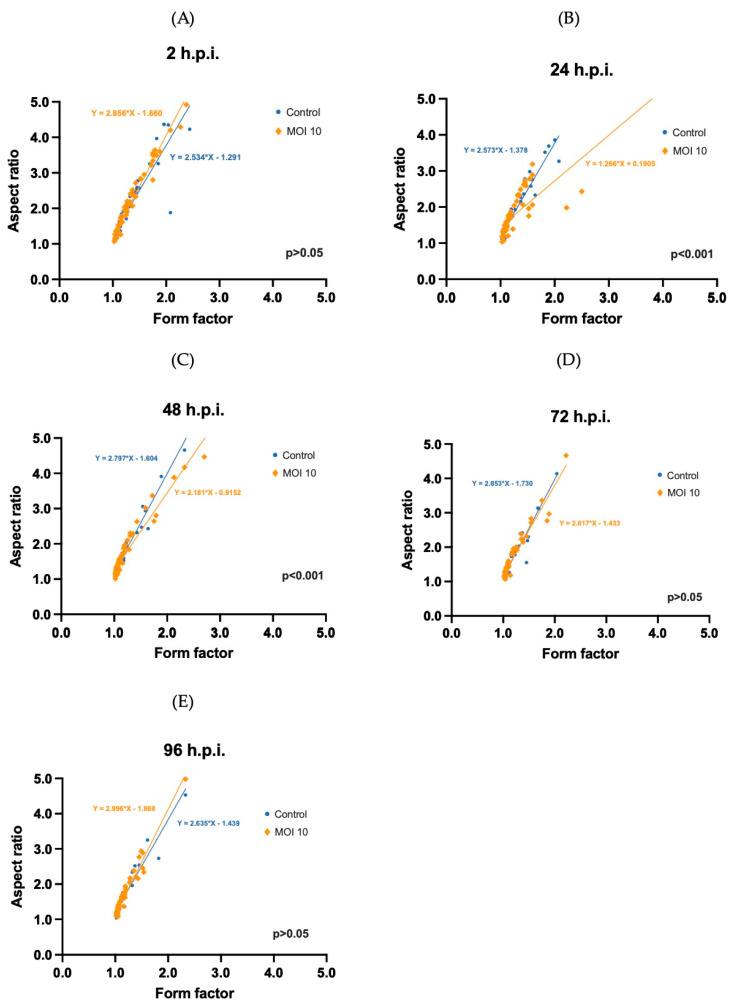
Mitochondrial complexity analysis. The complexity of mitochondria was evaluated by comparing the shape factor and aspect ratio at different times post infection (**A**–**E**). Control HEp-2 cells (*n* = 57) represented by the blue line, and HEp-2 cells infected with hRSV ON1 MOI10 (*n* = 57), represented by the orange line, were used. A two-sided Student’s *t*-test was performed to compare the equations of the post-infection timeline, and the *p*-values are shown on the graph.

**Figure 6 viruses-15-01518-f006:**
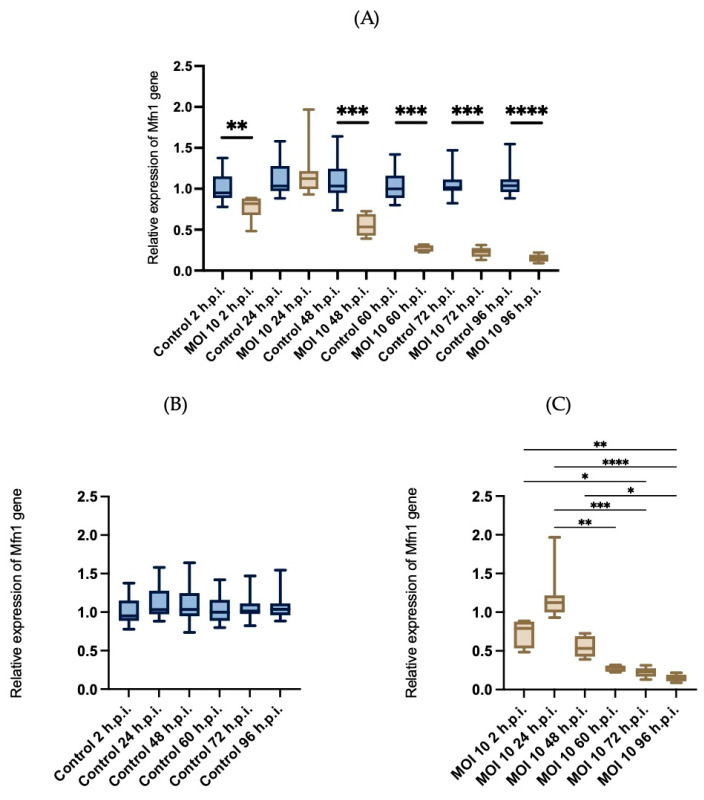
Evaluation of the relative expression of Mfn1 mRNA. The relative expression of the Mfn1 gene, normalized with GAPDH, was measured in control HEp-2 cells (blue boxes) andHEp-2 cells infected with ON1 respiratory syncytial virus (brown boxes). (**A**) Comparison of the relative expression of Mfn1 mRNA compared within time points, and within groups (**B**,**C**). Data are expressed as means and interquartile ranges; *n* = 6 per group. Friedman’s test was performed for each of the groups at different times post-infection, and statistically significant differences are indicated with an asterisk above the boxplots, * corresponds to *p* ≤ 0.03, ** corresponds to *p* ≤ 0.002, *** corresponds to *p* ≤ 0.0002 and **** corresponds to *p* ≤ 0.0001. A U-Mann Whitney test was performed for the comparison between the control and infected group at each of the time post infection The bars indicate the data between the lower quartile (25%) and the lower extreme and the upper quartiles (75%) and the upper extreme.

**Figure 7 viruses-15-01518-f007:**
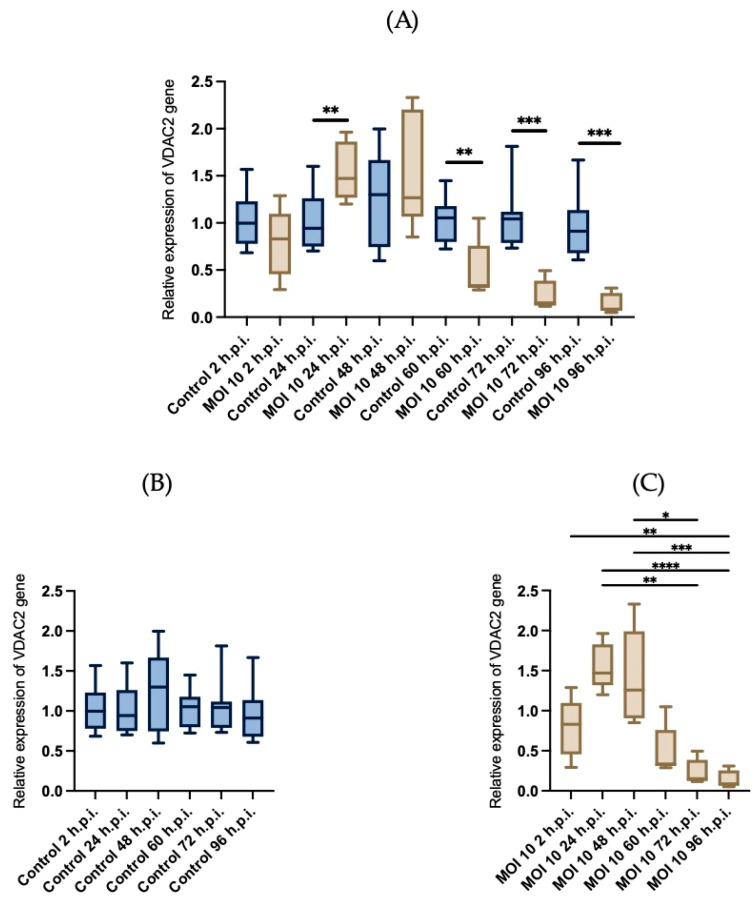
Evaluation of the relative expression of VDAC2 mRNAs. The relative expression of the VDAC2 gene, normalized with GAPDH, was in control HEp-2 cells (blue boxes) andHEp-2 cells infected with ON1 respiratory syncytial virus (brown boxes). (**A**) Comparison of the relative expression of Mfn1 mRNA compared within time points, and within groups (**B**,**C**). Data are expressed as means and interquartile ranges; *n* = 6 per group. Friedman’s test was performed for each of the groups at different times post-infection, and statistically significant differences are indicated with an asterisk above the boxplots, * corresponds to *p* ≤ 0.03, ** corresponds to *p* ≤ 0.002, *** corresponds to *p* ≤ 0.0002 and **** corresponds to *p* ≤ 0.0001. A U-Mann Whitney test was performed for the comparison between the control and infected group at each of the time post infection. The bars indicate the data between the lower quartile (25%) and the lower extreme and the upper quartiles (75%) and the upper extreme.

**Figure 8 viruses-15-01518-f008:**
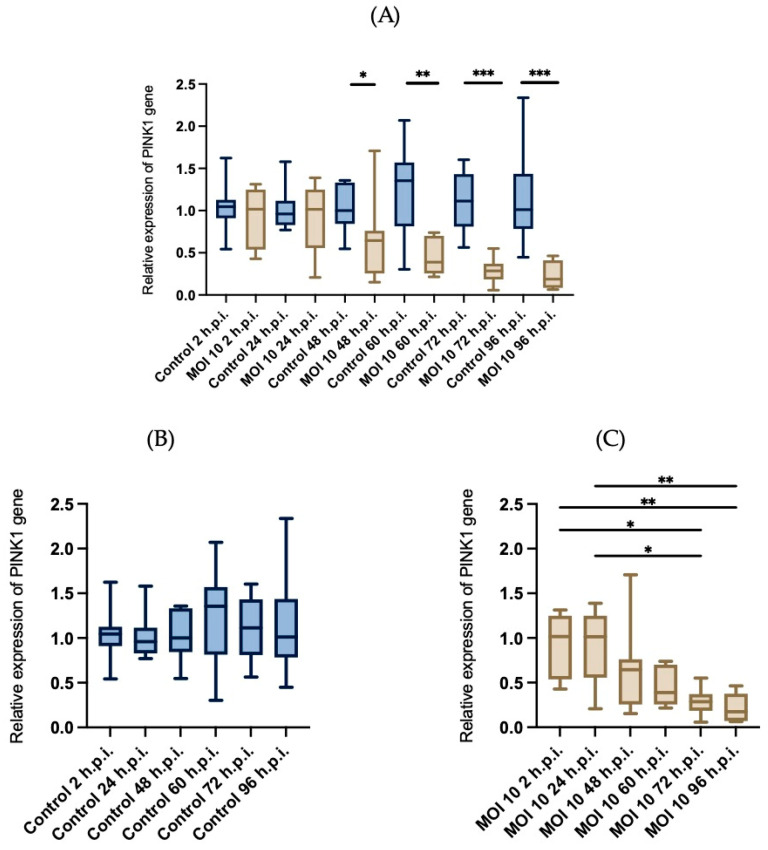
Evaluation of the relative expression of PINK1 mRNAs. The relative expression of the PINK1 gene, normalized with GAPDH, was measured was in control HEp-2 cells (blue boxes) andHEp-2 cells infected with ON1 respiratory syncytial virus (brown boxes). (**A**) Comparison of the relative expression of Mfn1 mRNA compared within time points, and within groups (**B**,**C**). Data are expressed as means and interquartile ranges; *n* = 6 per group. Friedman’s test was performed for each of the groups at different times post-infection, and no statistically significant differences was found. A U-Mann Whitney test was performed for the comparison between the control and infected group at each of the time post infection; statistically significant differences are indicated with an asterisk above the boxplots, * corresponds to *p* ≤ 0.03, ** corresponds to *p* ≤ 0.002, *** corresponds to *p* ≤ 0.0002. The bars indicate the data between the lower quartile (25%) and the lower extreme and the upper quartiles (75%) and the upper extreme.

**Table 1 viruses-15-01518-t001:** Design of primers for real-time PCR.

Gene	Primer’s Sequence (5′-3′)
PINK1 forward	GAGTATGGAGCAGTCACTTACAG
PINK1 reverse	CAGCACATCAGGGTAGTCG
Mfn1 forward	ACCTGTTTCTCCACTGAAGC
Mfn1 reverse	TGGCTATTCGATCAAGTTCCG
VDAC2 forward	TGTCTTTGGTTATGAGGGCTG
VDAC2 reverse	CCTCCAAATTCTGTCCCATCG
GAPDH forward	CAAGGCTGAGAACGGGAAGC
GAPDH reverse	AGGGGGCAGAGATGATGACC
MCVF	GGCAAATATGGAAACATACGTGAA
MCVR	TCTTTTTCTAGGACATTGTAYTGAACA

**Table 2 viruses-15-01518-t002:** Qualitative scoring system [40,41].

Injury Stage	Characteristics of Mitochondria
Class I	Mitochondria are considered normal. The cristae are densely packed and longitudinal, and the matrix is electron dense.
Class II	Mitochondria in this class have rounded or asymmetrical cristae. The cristae’s direction, narrowness, or regular distribution have been lost.
Class III	Mitochondria vary in size and form, with discontinuous membranes, fragmented cristae, and swollen and electro lucent matrix.
Class IV	In this class, the mitochondrial double membrane is no longer complete, and the cristae are disrupted and dispersed.

## Data Availability

All data are available upon request.

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
