# Peer review of "Ultrastructural and Functional Characterization of Mitochondrial Dynamics Induced by Human Respiratory Syncytial Virus Infection in HEp-2 Cells"

_viruses, 2023, doi:10.3390/v15071518_

Round 1

Reviewer 1 Report

In this work the authors shown that hRSV infection altars mitochondrial morphology trough the regulating of genes such as Mfn1, VDAC2, and PINK. The authors shown that hRSV, induces mitochondrial changes such as shortened cristae, swollen matrix, and damaged membrane. They showed as well the modulation of Mfn1, VDAC2, and PINK.

The work is interesting and tries to shed more light on the changes caused by infection with hRSV. There are more and more reports of the effect of viruses on different organelles such as mitochondria.

The article is well written, it is clear.

However I suggest several changes

Introduction.

It is well known that HRSV interacts with different cellular proteins, such as the  associated with the antiviral response. As well have been reported, changes in mitochondrial proteins and functions, cell cycle regulatory molecules, nuclear pore complex proteins and nucleus cytoplasmic trafficking proteins. The authors address properly those points.

 Results.

Figure 2 the author could explain why? the number of normal mitochondria is 100 in normal cells, then it goes down after 24 h and goes up again at 96 and goes down again, showing an erratic behavior. Class 2 shows a percentage of more than 50% at 72 hours in the controls and then drops again at 96 hours.

The authors must indicate how many independent experiments were performed.

They must indicate the SD of the number of experiments performed.

Figure 3

I suggest that shorten the scale, to make the changes more evident between the bars. The maximum scale could be 2.5. To contrast between the infected and control group, perhaps the best way to represent the results will be a single graph so that the comparison can be made between the infected and the control group. (Where are the mock, I understand that the control is the one only incubated with medium?’)

Why with a very big SD and with the data very close the authors show high significance.

Statistical comparison would have to be made between the two groups at the same time .

Is better to compare the size between the control and infected cells at different time

Same case for the figure 4

Why the authors did not measured the membrane potential, which is measured of the mitochondria damage.

Fiugure 6 if you have the title eliminate the Word control in axis x and eliminate moi of 10, because is in the title, but I consider that is better that you mix the Graphs     

Although is important to analyze the changes on the expression of proteins is crucial to compare between the control and the infected cells in each time

Once again I suggest to mix both graphics.

Line 513 references.

I consider it essential to have an infection control at least one WB or IF that shows that the infection is taking place. In any figure shown any evidence of infection.

On other hand I consider very important the evaluation of the expression of the gene Mfn, VDAC2 and PINK. However I consider that the data must be corroborated with the expression of the corresponding protein, since it is well known that the presence of the messenger does not necessarily correspond to the transcript due to the regulation.

Author Response

We appreciate all the comments, and we believe that we have answered all of them to the best of our knowledge. In fact, these comments have greatly improved our manuscript. We have highlighted in yellow all the changes and used track changes.

Introduction.

  1. It is well known that HRSV interacts with different cellular proteins, such as the  associated with the antiviral response. As well have been reported, changes in mitochondrial proteins and functions, cell cycle regulatory molecules, nuclear pore complex proteins and nucleus cytoplasmic trafficking proteins. The authors address properly those points.

We have extended the introduction by addressing this comment (lines 68-77) with the following text:

hRSV replication and transcription process occur in the cytoplasm and Golgi, but not in the nucleus (Cowton VM, J Virol, 2005; Ghidlyal R, J Gen Virol, 2005; Ghidlyal R, J Collins PL, 2013). Nonetheless, some viral proteins are imported into the nucleus. For example, the M2-1 viral protein promotes nuclei NF-kB activation and Rel-A (Reimers K, Virology, 2004), and the M protein interacts with zinc finger protein 2, SMAD3, and chromatin, inhibiting the host cell transcription (Li HM, J Virol, 2008; Li HM, Cells, 2021). NS1 and NS2 inhibit IRF-3, inhibit INF activation, and induce STAT2 degradation (Bossert B, J Virol, 2003; Elliot J, J Virol, 2007). Furthermore, it has been reported that hRSV antagonizes the innate immune response mediated by MDA5 and MAVS (Bhoj VG, PNAS, 2008; Lifland AW, 2012). Both proteins are part of the mitochondrial antiviral signaling system. In fact, hRSV interacts with the mitochondrial interface. This interaction impaired mitochondrial respiration caused loss of mitochondrial membrane potential, and increased mitochondrial ROS (Hu, M, Bichome Biophys Res Commun, 2017).

Results.

  1. Figure 2 the author could explain why? the number of normal mitochondria is 100 in normal cells, then it goes down after 24 h and goes up again at 96 and goes down again, showing an erratic behavior. Class 2 shows a percentage of more than 50% at 72 hours in the controls and then drops again at 96 hours.

The answer to this observation was added in lines 595-602

We observed that the percentage of class 2 mitochondria in the mock-infected cells increased at 24- and 72-h.p.i. It is important to point out that at time 0-h.p.i the cells were incubated with either virus (in PBS) or PBS, and after two hours, the inoculum was replaced with complete DMEM. Also, the media was replaced after 48-h.p.i. Therefore, it is likely that the replacement of PBS with complete MDEM and/or the media replacement process could stress the cells, resulting in an increase in the percentage of Class 2 mitochondria at 24 and 72 h.p.i. Nonetheless, we found that, independently from these changes, hRSV ON1 induced changes in mitochondrial morphology by inducing outer membrane discontinuities, altering the shape of mitochondrial cristae, and inducing the formation of swirling or fragmented cristae. In contrast, the mitochondrial morphology of hRSV-uninfected cells did not change over time (mitochondria had clearly defined cristae and membranes).

  1. The authors must indicate how many independent experiments were performed.

On the one hand for the ultra-structural analysis, each sample was done by duplicate, but there were no independent experiments. This is due to the laborious nature of thin-section TEM and the accessibility to TEM time. Nonetheless, in order to validate the mitochondrial damage observed by TEM we performed RT-qPCRs in duplicate with two biological replicas. We have addressed this issue in the legend of Figure 2 and in lines 167-168 and 219

  1. They must indicate the SD of the number of experiments performed.

Please see the previous answer. Because the TEM data was done only by duplicate, we did not calculate the Standard Deviation (as it will not be appropriated). Nonetheless, for the RT-qPCR data, we performed a boxplot graph that shows therefore the dispersion data as 25-75 percentiles and not at the Standard deviation.

  1. Figure 3

I suggest that shorten the scale, to make the changes more evident between the bars. The maximum scale could be 2.5. To contrast between the infected and control group, perhaps the best way to represent the results will be a single graph so that the comparison can be made between the infected and the control group. (Where are the mock, I understand that the control is the one only incubated with medium?’)

We were not able to change the scale to 2.5 as one point of the MO1 10 48 h.p.i. would be deleted. In the original manuscript the comparison between the mock-infected (the mock-infected control was inoculated PBS) and the infected cells. However, we have generated a new Figure 3 that shows in panel A the comparison between groups. The original figures 3A and 3B are now figures 3B and 3C respectively. We modified the text in section 3.3 to include this new figure.

  1. Why with a very big SD and with the data very close the authors show high significance.

Statistical comparison would have to be made between the two groups at the same time .

Is better to compare the size between the control and infected cells at different time

In line 249-251 we added:

The data in Figures 3, 4, 6, 7, and 8, is shown as a boxplot and the lines indicate the data between the lower quartile (25%) and the lower extreme, and between the upper quartile (75%) and the upper extreme.

We also added this clarification in the figure legends.

  1. Why the authors did not measured the membrane potential, which is measured of the mitochondria damage.

We acknowledge this suggestion; however, currently, we do not have the means nor the expertise to do these experiments. These experiments are beyond the scope of this article; here we aimed at describing the ultrastructural changes of the mitochondrion due to hRSV infection and to propose an initial mechanism. Nonetheless, in future work, we would like to define the exact mechanism and these experiments will then be extremely helpful.

  1. Figure 6 if you have the title eliminate the Word control in axis x and eliminate moi of 10, because is in the title, but I consider that is better that you mix the Graphs     

We have done this, and we have added a new panel that combines both graphs.

  1. Although is important to analyze the changes on the expression of proteins is crucial to compare between the control and the infected cells in each time

In the original manuscript, this information was found as supplementary information. Nonetheless, we included the suggested graph as a single panel in Figures 3, 4, 6, 7, and 8.

  1. Line 513 references.

I consider it essential to have an infection control at least one WB or IF that shows that the infection is taking place. In any figure shown any evidence of infection.

On other hand I consider very important the evaluation of the expression of the gene Mfn, VDAC2 and PINK. However I consider that the data must be corroborated with the expression of the corresponding protein, since it is well known that the presence of the messenger does not necessarily correspond to the transcript due to the regulation.

We agree with the reviewer that these experiments would be interesting. However, in the past 3 years, we have tried to import to Mexico these and other antibodies with little to no success. For some reason, the SAGARPA (the Mexican Ministry of Agriculture and Rural Development which is the entity that regulates these imports) has not allowed two different vendors to import these antibodies. Nonetheless, we observed downregulation for these three genes, hence it is unlikely that protein levels could be higher than in the mock-infected samples. In fact, if the case were to be the opposite (overexpression of these mRNAs) then we would need to perform Western blots or immunochemistry to be sure that also the protein levels would increase.

Reviewer 2 Report

General Comments:

Lara-Hernandez and colleagues report in this manuscript the ultrastructural and functional changes to mitochondria during human respiratory syncytial virus (hRSV) infection using thin-section transmission electron microscopy and RT-qPCR. Their findings suggest that hRSV-induced mitochondrial alterations involve shortened cristae, swollen matrix, and damaged membrane. Furthermore, the authors reveal that hRSV infection modifies mitochondrial morphodynamics by regulating key genes involved in the antiviral response, including Mfn1, VDAC2, and PINK1. Overall, this is interesting work that addresses a gap in our understanding of the hRSV lifecycle. The manuscript is well written, and I thought the Discussion section was particularly well done. I also noticed the following weaknesses that the authors should address:

Major Comments:

1. The MOI used in all experiments (MOI of 10) seems quite high. Most studies in the literature use an MOI of 0.1 to 3.0, including previous studies investigating the impact of RSV infection on mitochondria referenced in this manuscript (Ref 16, 17, 18, 23, 24, 25, 27). It is possible that viral load could have a direct impact on the observed morphological changes to mitochondria and changes to gene expression. The authors should consider repeating these experiments at a lower MOI for direct comparison to previous studies in the literature. Alternatively, this should be mentioned in the discussion section as a potential caveat of the study.

2.  RSV infected and control cells were grown in DMEM containing glucose, which allows for the utilization of both glycolysis and oxidative phosphorylation for ATP production. However, the nutrient composition of commercial media, like DMEM, can introduce metabolic artifacts in cancer cell lines, such as the HEp-2 cells used in this study (Cantor (2019) Trends Cell Biol 29:854–861, Cantor et al., (2017) Cell 169:258–272.e17, Vande Voorde et al., (2019) Sci Adv 5:eaau7314). To establish a more physiologically relevant condition, replacing glucose with galactose in the media can reprogram cultured cells from glycolysis to oxidative phosphorylation. This shift in metabolism can have implications for various cellular activities and physiology, including increased levels of ROS in cells with enhanced respiration, which can greatly affect host defenses. Furthermore, specific metabolites, serving as donors for enzymes involved in posttranslational modifications, can profoundly impact diverse cellular activities. To clarify the mechanisms and functional significance of mitochondrial interfaces during hRSV infection, it is advisable for the authors to replicate these experiments using a culture medium devoid of glucose (commercially available) and supplemented with galactose, forcing the cells to rely solely on oxidative phosphorylation for ATP generation. The authors should, at the very least, acknowledge this consideration in the discussion section.

3. It is unclear whether GAPDH is an appropriate endogenous housekeeping gene during hRSV infection for RT-qPCR analysis. A reference to a study demonstrating steady-state levels are not impacted by hRSV infection or metabolic perturbations (across the tested timepoints: 2, 24, 48, 60, 72, 96 hpi) should be included, or experiments demonstrating this should be included.

4. Supplemental figure S1 is not referenced within the main text. This figure demonstrates that the authors were unable to detect hRSV RNA at 72 and 96 hours post infection. It is therefore unclear whether the observed mitochondrial phenotypes at 72 and 96 hours post infection are due to the virus. This should be discussed.

5. Line 463: The authors conclude there are no significant differences in PINK1 expression within the control or infected groups but there is a trend to downregulate the expression of this gene in the infected group. However, Figure S6 indicates there is indeed a statistically significant reduction in gene expression in the infected group compared to the control at 48, 60, 72, and 96 hours post infection. This should be revised.

Minor Comments:

1. There is no mention of how hRSV ON1 stocks are produced, stored, or titrated.

2. Line 126: TEM has already been defined on line 119

3. Line 257: The arrow in Figure 1E is not defined in the legend.

4. Line 258: There are no arrows in Figure 1F. The “(arrows)” should be removed from this line.

5. Line 259: What do the asterisk indicate in Figure 1G?

6. Line 332: The words “cells” should be removed to read “performed in HEp-2 infected cells”

7. Line 352: the word “at” should be inserted between “groups most”

8. Line 422: the word difference is missing. It should read “statistically significant difference (p<0.001) between…”

9. Line 428: An extra 4 has been inserted.

10. Line 512-513: A reference should be included for this statement. What studying using confocal microscopy?

11. Line 514: MeV should have been defined earlier on line 78

12. Line 583: There is no need to shorten peritoneal macrophages to “(PM)” as it is not used again

13. There are a number of instances (i.e. lines 442, 454, 466, 585, Fig. S4 legend, Fig. S5 legend, and Fig. S6 legend) that refer to expression of the “protein”. The authors are using RT-qPCR, this is not a measure of protein expression but rather gene expression (RNA). This should be changed throughout the manuscript.

The quality of the English language was good, with only minor editing needed.

Author Response

We appreciate all the comments, and we believe that we have answered all of them to the best of our knowledge. In fact, these comments have greatly improved our manuscript. We have highlighted in yellow all the changes and used track changes.

Lara-Hernandez and colleagues report in this manuscript the ultrastructural and functional changes to mitochondria during human respiratory syncytial virus (hRSV) infection using thin-section transmission electron microscopy and RT-qPCR. Their findings suggest that hRSV-induced mitochondrial alterations involve shortened cristae, swollen matrix, and damaged membrane. Furthermore, the authors reveal that hRSV infection modifies mitochondrial morphodynamics by regulating key genes involved in the antiviral response, including Mfn1, VDAC2, and PINK1. Overall, this is interesting work that addresses a gap in our understanding of the hRSV lifecycle. The manuscript is well written, and I thought the Discussion section was particularly well done. I also noticed the following weaknesses that the authors should address:

Major Comments:

  1. The MOI used in all experiments (MOI of 10) seems quite high. Most studies in the literature use an MOI of 0.1 to 3.0, including previous studies investigating the impact of RSV infection on mitochondria referenced in this manuscript (Ref 16, 17, 18, 23, 24, 25, 27). It is possible that viral load could have a direct impact on the observed morphological changes to mitochondria and changes to gene expression. The authors should consider repeating these experiments at a lower MOI for direct comparison to previous studies in the literature. Alternatively, this should be mentioned in the discussion section as a potential caveat of the study.

We understand the point of the reviewer and we thank him as we have included this point in the discussion (Lines 608-616).

While most studies with hRSV use MOIs between 0.1 and 3.0 we decided to use an MOI of 10 to be able to detect ultrastructural changes, which is one of the limitations of this study. In previous studies with Zika (ZIKV) (10.1371/journal.pone.0283429) and Chikungunya (10.3390/v15010132) (CHIKV) virus, we used low MOIs for thin-section TEM. However, preliminary experiments with hRSV at MOIs of 0.1 and 1.0 showed that detecting ultrastructural changes in the mitochondrion was extremely challenging at these MOIs (data not shown). Furthermore, in the cases of CHIKV (10.3390/v15010132) and ZIKV (10.1371/journal.pone.0283429), infections at an MOI of 1.0 lead to complete destruction of the cellular monolayer within the first 72 h.p.i.; however, with hRSV, infection at an MOI of 10 does not result in complete disruption of the monolayer even at 96-h.p.i. and hence, we sought to use a high MOI to increase the chance of finding ultrastructural changes.

  1.  RSV infected and control cells were grown in DMEM containing glucose, which allows for the utilization of both glycolysis and oxidative phosphorylation for ATP production. However, the nutrient composition of commercial media, like DMEM, can introduce metabolic artifacts in cancer cell lines, such as the HEp-2 cells used in this study (Cantor (2019) Trends Cell Biol 29:854–861, Cantor et al., (2017) Cell 169:258–272.e17, Vande Voorde et al., (2019) Sci Adv 5:eaau7314). To establish a more physiologically relevant condition, replacing glucose with galactose in the media can reprogram cultured cells from glycolysis to oxidative phosphorylation. This shift in metabolism can have implications for various cellular activities and physiology, including increased levels of ROS in cells with enhanced respiration, which can greatly affect host defenses. Furthermore, specific metabolites, serving as donors for enzymes involved in posttranslational modifications, can profoundly impact diverse cellular activities. To clarify the mechanisms and functional significance of mitochondrial interfaces during hRSV infection, it is advisable for the authors to replicate these experiments using a culture medium devoid of glucose (commercially available) and supplemented with galactose, forcing the cells to rely solely on oxidative phosphorylation for ATP generation. The authors should, at the very least, acknowledge this consideration in the discussion section.

We understand the point of the reviewer. However, we chose this media because our group has been working with hRSV for a long time and we wanted to keep our study consistent with a previous study (10.1111/cei.12793). Furthermore, we also chose DMEM based on hRSV and Hep-2 cells literature (10.1016/j.virol.2009.03.008, 10.1016/j.jviromet.2004.02.020, 10.1371/journal.ppat.1009469, 10.1371/journal.pone.0029386, 10.1128/jvi.00215-12, and 10.1128/jvi.76.9.4287-4293.2002). We have cited these articles in section 2.1. We appreciate this suggestion, but the aim was to the effect on the mitochondrion between infected vs mock-infected cells. However, we will take into consideration this suggestion for future functional studies.

  1. It is unclear whether GAPDH is an appropriate endogenous housekeeping gene during hRSV infection for RT-qPCR analysis. A reference to a study demonstrating steady-state levels are not impacted by hRSV infection or metabolic perturbations (across the tested timepoints: 2, 24, 48, 60, 72, 96 hpi) should be included, or experiments demonstrating this should be included.

Please see lines 213-218 and supplementary figure S1

We decided to use GAPDH as a constitutive gene based on other studies with hRSV (10.1128/mbio.03528-21 and 10.1093/infdis/jix070). Furthermore, based on the method from Livak and Schmittgen 2001 we found no statically significant differences in the expression of GDAPH in the infected cells with respect to the mock-infected cells. We have included these two references in section 2.

  1. Supplemental figure S1 is not referenced within the main text. This figure demonstrates that the authors were unable to detect hRSV RNA at 72 and 96 hours post infection. It is therefore unclear whether the observed mitochondrial phenotypes at 72 and 96 hours post infection are due to the virus. This should be discussed.

We have clarified this point on lines (730-746).

On the one hand, most of the ultrastructural changes were only noticeable in early times, before there was an exponential growth in viral RNA (see Supplementary Figure 2). This could be due to the expression of early viral genes. At this stage, it is unclear which genes are involved in these changes as there is a debate about whether hRSV gene expression follows a gradient or a steady-state regime (10.1128/jvi.00102-07, 10.1371/journal.pone.0227558). Also, it seems to be a slight up-regulation of VDAC2 and Mnf1 only at early times, correlating almost perfectly with the alterations seen in Figure 5, before these genes are strongly down-regulated. On the other hand, at time points close and before the maximum viremia, most mitochondria have a normal morphology. This suggests that the downregulation of VDAC2, Mfn1, and PINK could be an indirect consequence of virus-induced cellular death, thus we were not able to observe abnormal mitochondria at these time points. It is important to point out that even at the peak of the viremia, we were not able to observe the complete destruction of the monolayer. Therefore, the lack of monolayer destruction and the abrupt drop in viremia at 72-h.p.i. suggest that the downregulation of VDAC2, Mfn1, and PINK might be an antiviral response that limits the progression of the infection, and therefore at these time points most mitochondria have normal morphology.

  1. Line 463: The authors conclude there are no significant differences in PINK1 expression within the control or infected groups but there is a trend to downregulate the expression of this gene in the infected group. However, Figure S6 indicates there is indeed a statistically significant reduction in gene expression in the infected group compared to the control at 48, 60, 72, and 96 hours post infection. This should be revised.

We thank the reviewer for this observation, we were not careful enough and we have replaced that sentence (lines 540-542):

In the case of the expression of PINK1 mRNA (figure 8), there are no significant differences in the mock-infected cells at different time points. However, in hRSV-infected cells, the expression of this mRNA is downregulated, especially at 48-, 60-, 72-, and 96 h.p.i.

Minor Comments:

  1. There is no mention of how hRSV ON1 stocks are produced, stored, or titrated.

We have added this text in lines 120-128 section 2.1

The viral stock was generated from a clinical sample positive for hRSV, the genotype was assigned by sequencing the ectodomain region of the G gene, and the isolated was used subsequently to infect confluent monolayer HEp-2 cells. The presence of the cytopathic effect of the virus was identified (by light microscopy after five days, and the presence of the virus was confirmed by qRT-PCR (with absolute quantification of the number of viral copies); a subculture was performed at MOI 5, and virus concentration was carried out at low speed starting from 25.2 x106 infected cells at MOI 5 based on the protocol of Rayaprolu et al. The viral stock was quantified by qRT-PCR using the primers in Table 2 and stored for less than six months refrigerated at 4°C.

  1. Line 126: TEM has already been defined on line 119

Done

  1. Line 257: The arrow in Figure 1E is not defined in the legend.

Done

  1. Line 258: There are no arrows in Figure 1F. The “(arrows)” should be removed from this line.

Done

  1. Line 259: What do the asterisk indicate in Figure 1G?

Done

  1. Line 332: The words “cells” should be removed to read “performed in HEp-2 infected cells”

Done

  1. Line 352: the word “at” should be inserted between “groups most”

Done

  1. Line 422: the word difference is missing. It should read “statistically significant difference (p<0.001) between…”

Done

  1. Line 428: An extra 4 has been inserted.

Done

  1. Line 512-513: A reference should be included for this statement. What studying using confocal microscopy?

Done

  1. Line 514: MeV should have been defined earlier on line 78

Done

  1. Line 583: There is no need to shorten peritoneal macrophages to “(PM)” as it is not used again

Done

  1. There are a number of instances (i.e. lines 442, 454, 466, 585, Fig. S4 legend, Fig. S5 legend, and Fig. S6 legend) that refer to expression of the “protein”. The authors are using RT-qPCR, this is not a measure of protein expression but rather gene expression (RNA). This should be changed throughout the manuscript.

Thanks, this was an oversight and we have fixed

Round 2

Reviewer 1 Report

Thank you for the effort in approaching and considerably improving this article.

I am satisfied with the work that was done, all the points and suggestions that I sent were addressed appropriately.